# Contrast-enhanced ultrasound for the diagnosis of acute pyelonephritis in pediatric patients with urinary tract infection: A feasibility study

**Han Byeol Lee[1], Seunghyun Lee[1,2]\*, Young Hun Choi[1,2], Jung-Eun Cheon[1,2,3], Seul Bi Lee[1], Yeon Jin Cho[1], Yo Han Ahn[4,5], Seon Hee Lim[6]**

1 Department of Radiology, Seoul National University Children's Hospital, Seoul, Republic of Korea,
2 Department of Radiology, Seoul National University College of Medicine, Seoul, Republic of Korea,
3 Institute of Radiation Medicine, Seoul National University Medical Research Center, Seoul, Republic of Korea, 4 Department of Pediatrics, Seoul National University Children's Hospital, Seoul, Republic of Korea,
5 Department of Pediatrics, Seoul National University College of Medicine, Seoul, Republic of Korea,
6 Department of Pediatrics, Uijeongbu Eulji Medical Center, Uijeongbu, Kyungki-do, Republic of Korea

\* seunghyun.lee.22@gmail.com

**Data Availability Statement:** All relevant data were registered in Figshare after removing patient information including sex, date of birth and study

## Abstract

### Purpose

This study aimed to assess the feasibility of contrast-enhanced ultrasound (CEUS) for the diagnosis of acute pyelonephritis (APN) in pediatric patients with febrile urinary tract infection (UTI).

### Materials and methods

Between March 2019 and January 2021, study participants with suspected UTI were assessed for APN using ultrasound. Parenchymal echogenicity changes, renal pelvis dilatation, and the presence of a focal suspected lesion were assessed using conventional grayscale ultrasound. The presence and location of a decreased perfusion area were evaluated using color Doppler ultrasound (CDUS) and CEUS. Agreement between each ultrasound examination and a $^{99m}$Tc–dimercaptosuccinic acid (DMSA) scan was assessed using the κ value, and the most visible period of the lesion was evaluated using CEUS.

### Results

This study enrolled 21 participants (median age, 8.0 months; range, 2.0–61.0 months) with isolated urinary tract pathogens. Five increased parenchymal echotextures (11.9%) and 14 renal pelvic dilatations (33.3%) were confirmed, but no focal lesions were detected on the grayscale images. CDUS and CEUS showed decreased local perfusion suggestive of APN in two and five kidneys, respectively. DMSA scan showed substantial agreement with CEUS findings (κ = 0.80, P = 0.010), but other grayscale and CDUS findings did not agree with DMSA scan results (P > 0.05). All lesions were best observed in the late parenchymal phase on CEUS.

date. (DOI: https://doi.org/10.6084/m9.figshare.21947219.v1).

**Funding:** CYH, a research grant from Samsung Medison Co., Ltd. (No. 0620192390, https://www.samsungmedison.com/) The funders had no role in study design, data collection and analysis, decision to publish, or preparation of the manuscript.

**Competing interests:** The authors have declared that no competing interests exist.

## Conclusion

CEUS can reveal renal perfusion defects in pediatric patients with suspected APN without radiation exposure or sedation; therefore, CEUS may be a feasible and valuable diagnostic technique.

## Introduction

Acute pyelonephritis (APN) requires appropriate diagnosis and treatment to prevent complications such as irreversible scarring, hypertension, chronic kidney disease, and sepsis in pediatric patients [1, 2]. However, clinical and laboratory findings are not always specific enough to confirm the diagnosis of APN, especially in infants with febrile urinary tract infection (UTI) [3]. Therefore, imaging modalities, such as ultrasonography, magnetic resonance imaging, $^{99m}$Tc–dimercaptosuccinic acid (DMSA) scan, or contrast-enhanced computed tomography (CECT) may be needed to assess the involvement of the kidney in UTI.

Ultrasonography is the first choice among imaging modalities to assess parenchymal abnormalities of the kidneys without radiation exposure [4]. However, grayscale ultrasound images are challenging to interpret and require the efforts of experienced operators to assess APN [4, 5]. To overcome the limitations of conventional grayscale ultrasonography, color Doppler ultrasound (CDUS) can help detect the hypoperfusion area. However, substantial cooperation is also needed for obtaining reliable findings while examining irritable young infants [4, 5]. CECT and DMSA scan are not routinely recommended for the initial workup process in pediatric patients because of radiation exposure and the need for sedation [6].

Contrast-enhanced ultrasound (CEUS) may be an excellent alternative for diagnosing APN in pediatric febrile UTI patients. CEUS can show perfusion abnormalities in the renal parenchyma by assessing the absence or decrease of enhancement after injection of microbubble contrast agents [7]. However, many countries still do not allow the use of ultrasonography contrast agents in pediatric patients. Therefore, studies related to CEUS in patients with APN are currently reported in case series with off-label use.

In this study, we prospectively recruited participants with febrile UTI; performed grayscale ultrasound, CDUS, and CEUS to detect renal parenchymal changes; and finally performed DMSA scans if focal renal lesions were found, APN related to vesicoureteral reflux (VUR) was suspected, or if the patient had a history of APN. Thus, this study was aimed at assessing the feasibility and usefulness of CEUS in diagnosing APN in pediatric patients with febrile UTI.

## Materials and methods

The Institutional Review Board of Seoul Nation University Hospital approved this study (H-1806-091-951) and written informed consent was obtained from the parents of subjects. Our single-institutional and prospective study was performed according to the Declaration of Helsinki, and the Institutional Review Board of our hospital approved this study (H-1806-091-951). Since the use of intravenous ultrasound contrast agents is not yet permitted in domestic pediatric patients, we prospectively recruited participants after obtaining approval for clinical trials from the Ministry of Food and Drug Safety. Written informed consent was obtained from the parents of subjects hospitalized for suspected UTI, along with an explanation of the study.

## Participants

Between March 2019 and January 2021, study participants were recruited via the emergency clinics of the Department of Pediatric Nephrology in our hospital according to the following eligibility criteria: the patient (a) had clinically suspected UTI, (b) was scheduled to undergo ultrasonography for the evaluation of renal parenchyma, and (c) was aged between two months and seven years. Exclusion criteria were as follows: the patient (a) had parents or legal representatives who could not understand explanations due to old age or intellectual disabilities, and (b) had any contraindication to CEUS, including hypersensitivity to sulfa-hexafluoride or any other component of this contrast agent, kidney dysfunction (estimated glomerular filtration rate $< 30$ mL/min per 1.73 m$^2$), right-to-left shunts, severe pulmonary hypertension (pulmonary artery pressure >90 mmHg), or uncontrolled systemic hypertension [8–10]. A total of 72 patients with suspected febrile urinary tract infection underwent a renal ultrasound during the study period, of which 21 patients participated in this study. Sex, age, clinical symptoms such as fever, fever duration, and total number of episodes were recorded at enrollment. Early study termination was considered if hypersensitivity to the contrast agent developed during the initial injection. The patient's pulse rate and oxygen saturation were monitored while the ultrasound contrast medium was injected, and blood pressure and oxygen saturation were monitored for 30 minutes after the end of the test. In addition, vital signs were additionally monitored for 30 minutes after transfer to the ward. Clinical characteristics, including white blood cell counts, C-reactive protein levels, and identified urinary pathogens were recorded based on laboratory findings.

## Image acquisition

All patients were requested not to fast before the CEUS examination. Renal ultrasonography examinations were conducted using one commercially available ultrasound scanner (RS85; Samsung Medison, Korea) and a pediatric convex transducer (CA3-10A) by two experienced pediatric radiologists, with 13 and 7 years of experience in pediatric ultrasonography examinations, respectively. No sedatives or other invasive interventions were administered, but the patients were calmed down by feeding them milk during the ultrasonography examinations.

First, a conventional grayscale image scan was used to assess renal parenchymal echogenicity changes, renal pelvis dilatation and wall thickening, distal ureter dilatation, and bladder wall thickening. CDUS examination was performed to evaluate the presence of a decreased perfusion area in the renal parenchyma.

CEUS was conducted with the following parameters: mechanical index, 0.069; gain, 50%–70%; dynamic range, 70; and frame rate, 9–12 images per second. An ultrasound contrast agent (SonoVue; Bracco, Italy) was injected through a peripheral intravenous injection. According to the pediatric CEUS guidelines, patients were administered 0.03 mL/kg body weight of contrast agent for each injection, immediately followed by 5 mL of normal saline flushing [9]. CEUS examination lasted until most of the bubbles had disappeared from the kidney, and data were stored (divided into 90 seconds clips) on hard disc. Bilateral kidneys were examined, and images were obtained from the longitudinal and axial planes. After completion of CEUS examination of one kidney, there was an interval of approximately 20 min before the first injected contrast agent was eliminated through the lungs. The contralateral kidney was examined after injecting the same volume of the ultrasound contrast agent. The presence and location of decreased perfusion were recorded, and after CEUS, the patients were monitored for 30 minutes to observe for any side effects.

Finally, the DMSA scan performed at the nearest time in patients with focal renal lesions on grayscale ultrasound, CDUS, or CEUS was used as the reference standard examination. In

addition, even if there were no focal renal lesions on ultrasound, DMSA scan was performed if APN related to VUR was suspected or if the patient had a history of APN. APN related to VUR was suspected when distal ureter dilatation was observed on grayscale ultrasound and voiding cystourethrography showed VUR of grade 3 or higher [11]. The DMSA scan, which included anterior, posterior, and both posterior oblique images, was performed 2 h after intravenous injection of $^{99m}$Tc-labeled DMSA using a tomographic gamma camera (E.cam signature, Siemens, Erlangen, Germany). During the DMSA scan, all patients were sedated by oral administration of 50 mg/kg body weight of chloral hydrate (Pocral Syrup; Hanlim, Korea) 30 minutes before the examination.

## Data analysis

Two certified radiologists, with 13 and 7 years of experience, respectively, in pediatric ultrasonography examinations reviewed the grayscale, CDUS, and CEUS images from the renal ultrasonography examinations. The reviewers assessed the following features with consensus: (1) change in renal parenchymal echogenicity; (2) renal pelvic dilatation and wall thickening; (3) associated findings such as distal ureter dilatation and bladder wall thickening; (4) suspected focal APN lesion; (5) area of hypoperfusion on CDUS; and (6) an area of hypoperfusion on CEUS. DMSA scans were used to verify APN lesions when APN was suspected on CEUS by recording the lesion site (right or left) and location (upper pole, mid-portion, or lower pole). DMSA scan results were interpreted independently by a pediatric radiologist who was blinded to the results of the other imaging studies prior to interpretation. The same analysis of follow-up grayscale images was performed if the follow-up ultrasound was performed instead of a DMSA scan. When a localized perfusion decrease was observed on CEUS, the best viewing period was also evaluated with respect to the time after contrast agent injection and enhancement pattern of the kidney. The enhancement phases were divided into a cortical phase in which renal cortical enhancement was seen and a parenchymal phase in which both cortical and medullary enhancement were visible. The parenchymal phase was separated into early and late parenchymal phases for practical reasons, such as the capacity to store images [12].

## Statistical analysis

Descriptive data were expressed as median values (range) for continuous variables and frequencies with percentages for categorical variables. Agreement between each ultrasound examination and DMSA scan was expressed as a κ value to account for the chance of agreement. Degree of agreement was categorized as follows: poor if $\kappa < 0.00$, slight if $0.00 \leq \kappa \leq 0.20$, fair if $0.21 \leq \kappa \leq 0.40$, moderate if $0.41 \leq \kappa \leq 0.60$, substantial if $0.61 \leq \kappa \leq 0.80$, and almost perfect if $\kappa > 0.80$. Statistical analyses were performed using SPSS version 25.0, for Windows (IBM Corp., Armonk, NY, USA). Statistical significance was set at a $P$-value $<0.05$.

## Results

Table 1 summarizes the clinical characteristics of the participants. This study enrolled 21 infants and children (median age, 8.0 months; range, 2.0–61.0 months) with suspected UTI. Fever lasted for a median of 2.0 days, and 57.1% of the patients had two or more UTI episodes in the past. The laboratory findings showed serum samples with a wide range of white blood cell counts (median, 14.4×10$^3$; range, 2.9–33.1×10$^3$ /μL) and C-reactive protein levels (median, 4.5; range 0.5–14.5 mg/dL). All patients had pyuria, and the most common isolated urinary tract pathogens were *Escherichia coli* (61.9%).

Table 2 shows the renal ultrasound results of 42 kidneys of 21 patients with suspected UTI. Grayscale ultrasonography findings showed five increased parenchymal echotextures (11.9%)

**Table 1. Clinical characteristics (N = 21).**

| Characteristics | Values |
|---|---|
| Age (months) | 8.0 (2.0–61.0) |
| Sex (boys:girls) | 13:8 |
| Clinical symptoms | |
| Fever (°C) | 38.8 (38.1–40.4) |
| Fever duration (days) | 2.0 (1.0–4.0) |
| Total number of UTI episodes (number, %) | |
| 1 | 9 (42.9%) |
| 2 | 8 (38.1%) |
| 3 | 4 (19.0%) |
| Laboratory findings | |
| WBC (×10³ /μL) in serum | 14.4 (2.9–33.1) |
| C-reactive protein (mg/dL) | 4.5 (0.5–14.5) |
| Identified pathogens in urine (number, %) | |
| *Escherichia coli* | 13 (61.9%) |
| Others* | 5 (23.8%) |
| No growth | 3 (14.3%) |

Note—All continuous values are presented as median values (range). *Others include *Citrobacter freundii* (n = 2), *Enterococcus faecalis* (n = 2), and *Klebsiella aerogenes* (n = 1). UTI = urinary tract infection. WBC = white blood cell count.

and 14 renal pelvic dilatations (33.3%), but no focal lesions were detected. However, CDUS and CEUS showed area of hypoperfusion suggestive of APN in two and five kidneys, respectively. Two kidneys that showed area of hypoperfusion on CDUS showed decreased local perfusion at the same location on CEUS.

DMSA scan was performed on 10 kidneys of five patients (**Table 3**). The mean time from ultrasound examination to DMSA scan was 137.2 ± 166.6 days. DMSA scan showed excellent agreement with CEUS findings (κ = 0.80, $P = 0.010$), but other grayscale and CDUS findings did not significantly agree with DMSA scan results ($P > 0.05$).

**Table 4** summarizes the test results of the 5 patients who underwent DMSA scan. There were several discordant findings between ultrasound examinations and DMSA scan. The

**Table 2. Renal ultrasonography in clinically suspected UTI (N = 42).**

| Findings | Number of kidneys (%) |
|---|---|
| Grayscale | |
| Echotexture Changes | 5 (11.9) |
| Renal pelvic dilatation | 14 (33.3) |
| Distal ureter dilatation | 7 (16.7) |
| Bladder wall thickening | 11 (26.2) |
| Focal lesion suspected APN | 0 (0.0) |
| Normal | 13 (31.0) |
| CDUS | |
| Perfusion decreased area | 2 (4.8) |
| CEUS | |
| Perfusion decreased area | 5 (11.9) |

Note—APN = acute pyelonephritis. CDUS = color Doppler ultrasound. CEUS = contrast-enhanced ultrasonography.

**Table 3. Comparison of ultrasound images and DMSA scan results (N = 10)*.**

| Parameters | DMSA Scan | | $\kappa^{\dagger}$ | P-value |
|---|---|---|---|---|
| | Positive (N = 5) | Negative (N = 5) | | |
| Grayscale | | | | |
| Echotexture changes | 2 | 0 | 0.40 | 0.114 |
| Renal pelvic dilatation | 4 | 1 | 0.60 | 0.058 |
| Distal ureter dilatation | 2 | 1 | 0.20 | 0.490 |
| Bladder wall thickening | 2 | 2 | 0.00 | 1.000 |
| Focal lesion suspected APN | 0 | 0 | 0.00 | |
| CDUS | | | | |
| Perfusion decreased area | 2 | 0 | 0.40 | 0.114 |
| CEUS | | | | |
| Perfusion decreased area | 4 | 0 | 0.80 | **0.010** |

Note—*Of the 42 kidneys suspected of having UTI, a DMSA scan was performed for a total of 10 kidneys that showed local lesions on ultrasonography to check for abnormalities.

†κ value for agreement between the results using kappa statistics. Almost perfect agreement, κ > 0.80; substantial agreement, $0.61 \leq \kappa \leq 0.80$; moderate agreement, $0.41 \leq \kappa \leq 0.60$; fair agreement, $0.21 \leq \kappa \leq 0.40$; slight agreement, $0.00 \leq \kappa \leq 0.20$; poor agreement, κ < 0.00. DMSA = [99m]Tc–dimercaptosuccinic acid. APN = acute pyelonephritis. CDUS = color Doppler ultrasound. CEUS = contrast-enhanced ultrasound.

CEUS of patient no. 1 showed bilateral renal multifocal perfusion defects at the locations corresponding to DMSA findings, but grayscale ultrasound did not detect any lesions in either of the two kidneys, and CDUS did not detect the right kidney lesion (Fig 1). The CEUS of patient no. 7 showed additional mid- and lower-polar perfusion defects at the locations corresponding to DMSA findings, but grayscale and CDUS could not detect mid- and lower-polar lesions in the right kidney (Fig 2). In addition, there were multifocal cortical defects on the DMSA scan of patient no. 20, but all ultrasounds, including CEUS, revealed no abnormal findings (Fig 3). Despite the normal-looking ultrasonographic findings, a DMSA scan was performed in this case because the patient had a history of APN. The CEUS findings did not identify the lesion because of poor lesion visibility during the early parenchymal phase. A visible lesion was detected only in the late parenchymal stage on a retrospective review after a DMSA scan. Patient no. 16 underwent a DMSA scan because APN related to VUR was suspected, but there were no abnormal findings in the ultrasound and DMSA scan results.

**Table 4. Test results of the 5 patients who underwent DMSA scan.**

| No. | Location | Grayscale | | | CDUS | CEUS | DMSA |
|---|---|---|---|---|---|---|---|
| | | Echotexture Changes | Renal pelvic dilatation | Focal lesion suspected APN | | | |
| 1 | Left | Y | Y | N | Y (Upper/Lower) | Y (Upper/Lower) | Y (Upper/Lower) |
| | Right | Y | Y | N | N | Y (Lower) | Y (Lower) |
| 7 | Left | N | N | N | N | N | N |
| | Right | N | Y | N | Y (Upper) | Y (Upper/Mid/Lower) | Y (Upper/Mid/Lower) |
| 12 | Left | N | N | N | N | N | N |
| | Right | N | Y | N | N | Y (Upper/Mid/Lower) | Y (Upper/Mid/Lower) |
| 16 | Left | N | Y | N | N | N | N |
| | Right | N | N | N | N | N | N |
| 20 | Left | N | N | N | N | N | Y (Upper/Lower) |
| | Right | N | N | N | N | N | N |

Note—DMSA = [99m]Tc–dimercaptosuccinic acid. APN = acute pyelonephritis. CDUS = color Doppler ultrasound. CEUS = contrast-enhanced ultrasound.

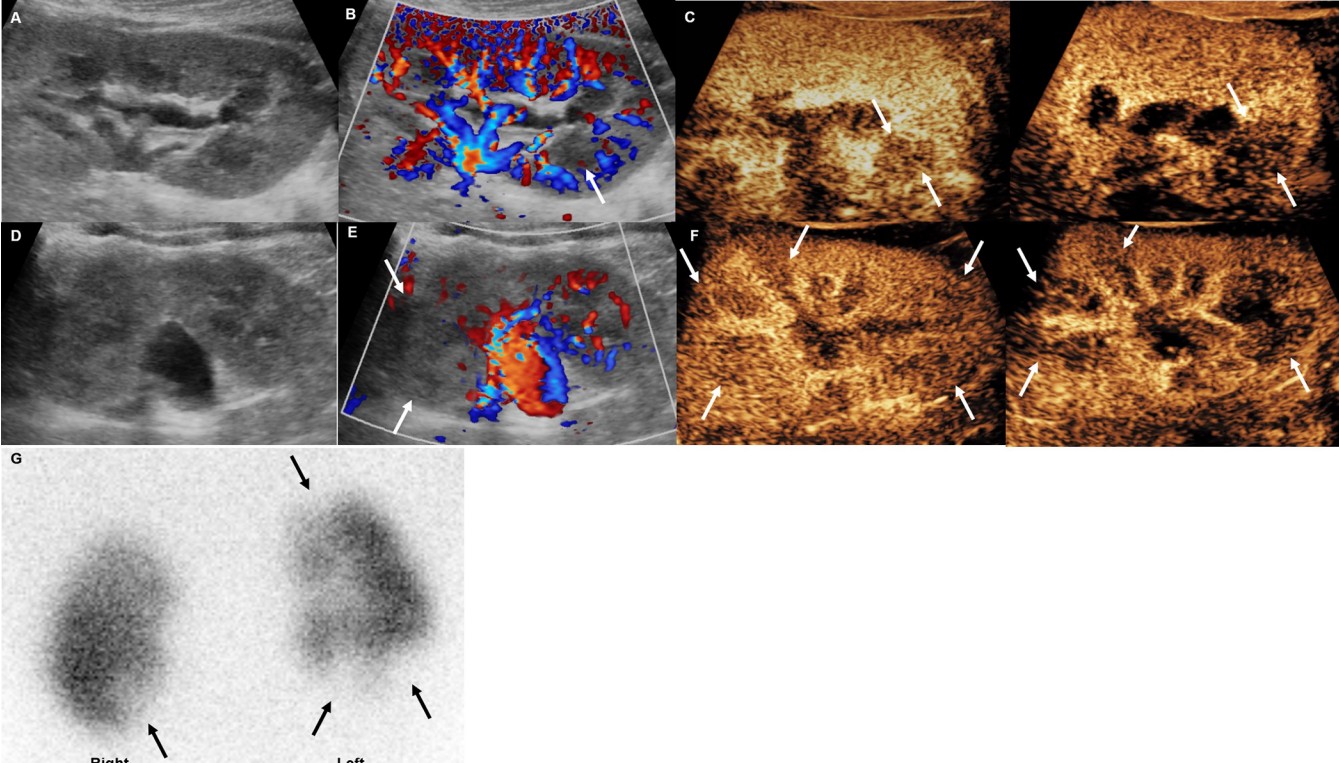

**Fig 1. Febrile urinary tract infection in a 20-month-old boy (patient no. 1).** (A) The grayscale ultrasound examination revealed increased cortical echotexture but no demonstrable focal lesion in the right kidney. (B) The hypoperfusion area at the lower polar area, which was seen on contrast-enhanced ultrasound (CEUS), was missed by color Doppler ultrasound (CDUS) (arrow). (C) CEUS showed definite hypoperfusion area (arrows) in the right kidney lower pole in the cortical phase (left) and early parenchymal stage (right). (D, E) The left kidney also showed increased cortical echotexture and focal hypoperfusion area (arrows) at the upper pole on CDUS. (F) CEUS showed multifocal hypoperfusion area (arrows) in the left kidney in the early parenchymal phase (left) and late parenchymal stage (right). (G) After two days, 99mTc–dimercaptosuccinic acid scan revealed multifocal cortical defects.

A total of six patients had a follow-up ultrasound instead of a DMSA scan after CEUS (490.0 ± 168.5 days), of which only one showed abnormal findings in the grayscale ultrasound, such as parenchymal atrophy. In this patient, the initial grayscale ultrasound showed diffusely increased parenchymal echotexture and slight pelvic dilatation of both kidneys, but no focal lesions were found on CEUS. A follow-up voiding cystourethrography in this patient revealed VUR, and the parenchymal damage was confirmed in the follow-up ultrasound. Therefore, this case could be regarded as a false-negative case of CEUS. The remaining five patients showed normal ultrasound findings, which indirectly confirmed that there was no APN at the time of CEUS. A follow-up ultrasound was also performed on three patients who underwent a DMSA scan, and a follow-up ultrasound confirmed the atrophy or cortical thinning remained in the area where cortical defects were observed in the DMSA scan.

In addition, five kidneys that showed perfusion defects on CEUS were analyzed to determine the optimal phase for lesion detection. The mean time to reach each phase was as follows: 16.3 ± 7.1 s to reach cortical phase, 84.3 ± 10.9 s to reach early parenchymal phase, and 127.5 ± 18.5 s to reach late parenchymal phase. The parenchymal phase was separated into early and late parenchymal phases for practical reasons, such as the capacity to store images [12]. No lesions were seen in the cortical phase; lesions were visible in three kidneys in the early parenchymal stage, but all lesions were best seen in the late parenchymal phase.

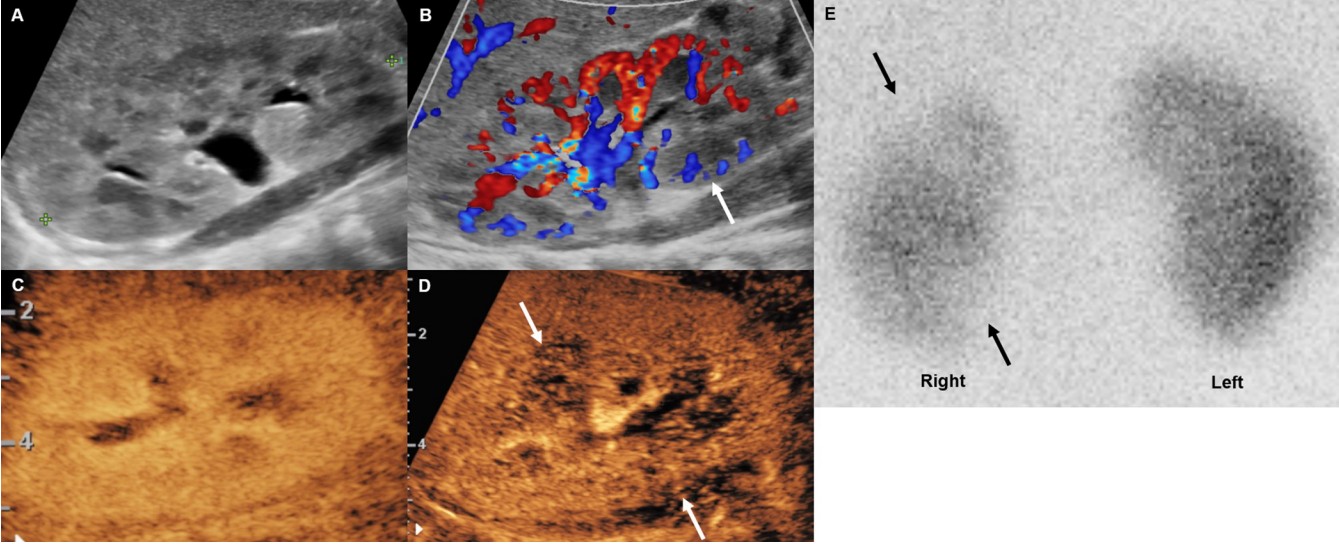

**Fig 2. Febrile urinary tract infection in a 2-month-old boy (patient no. 7).** (A) The grayscale ultrasound examination revealed no abnormalities in the right kidney. (B) The hypoperfusion area at the mid and lower polar area, which were seen on contrast-enhanced ultrasound (CEUS), was missed by color Doppler ultrasound (CDUS) (arrow). (C, D) CEUS showed no perfusion decrease in the right kidney in the cortical phase (C) but showed multifocal lesions in the late parenchymal stage (D). (E) After six days, 99mTc–dimercaptosuccinic acid scan revealed multifocal cortical defect.

## Discussion

Although DMSA and CECT scans are considered standard examination methods among several imaging techniques for evaluating renal involvement of UTIs, in practice, ultrasound is primarily used for renal evaluation in young pediatric UTI patients [3, 13]. Because additional imaging, such as DMSA, is usually performed when there is a local lesion in the kidney or evidence of decreased perfusion as a result of the ultrasound examination, efforts to increase the sensitivity of the renal ultrasound examination as an initial screening test are necessary.

Grayscale ultrasound can be safely and efficiently performed in pediatric patients, but it has low sensitivity with non-specific findings and high operator dependency for detecting parenchymal changes of the kidney [3]. Grayscale ultrasonography showed increased parenchymal echotextures and renal pelvic dilatations in pediatric patients with suspected APN, but no focal renal lesions were detected. Although CDUS can help detect reduced perfusion, it requires significant patient cooperation during the examination; therefore, perfusion-localized lesions are often missed in infants [4, 14]. CEUS can show a more objective hypoperfusion area and is less sensitive to the patient's motion, allowing a relatively accurate assessment of kidney perfusion [15]. Therefore, CEUS can be an excellent alternative for detecting renal parenchymal lesions in pediatric febrile UTI patients.

A previous retrospective study in pediatric patients showed that CEUS could accurately detect renal parenchymal lesions in children with febrile UTI [16]. The authors reported that the sensitivity and specificity of CEUS were 86.8% and 71.4%, respectively, using the DMSA scan as the reference standard. In another study of adult patients who underwent kidney transplantation, CEUS was able to accurately diagnose acute pyelonephritis [17]. Similar to these studies, the present study corroborates that CEUS shows excellent agreement with DMSA scan ($\kappa = 0.80$) for detection of renal parenchymal lesions.

Local hypoechoic areas seen on CEUS are thought to be due to reduced perfusion, inflammatory infiltration, edema, or capillary damage [7, 12, 18]. A previous report suggested that every phase should be carefully analyzed, as focal pyelonephritis is most prominent during the

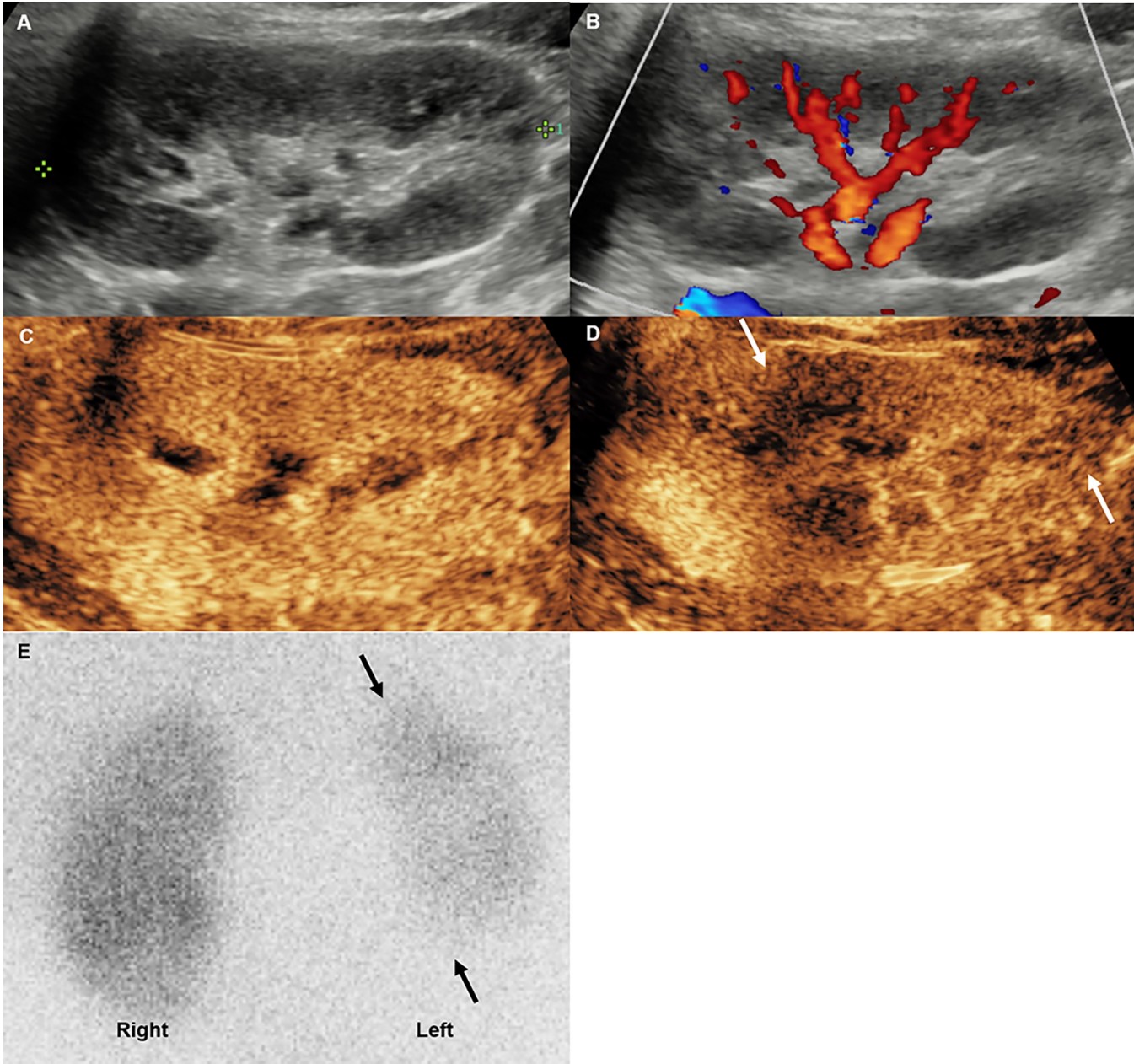

**Fig 3. Febrile urinary tract infection in a 38-month-old girl (patient no. 20).** (A) The grayscale ultrasound examination revealed no abnormalities in the left kidney. (B) Color Doppler ultrasound examination also did not reveal the focal perfusion abnormality. (C) CEUS showed no hypoperfusion area in the left kidney in the early parenchymal phase but showed a suspicious decreased renal enhancement, and (D) showed multifocal lesions in the late parenchymal stage. (E) After three days, 99mTc–dimercaptosuccinic acid scan revealed multifocal cortical defects in the left kidney.

late parenchyma phase [12]. Lesion visibility could be assessed during contrast injection of the CEUS, and in our study, almost all lesions were best observed in the late parenchymal phase. However, there might be an unsolved problem with regard to diffuse renal parenchymal involvement in APN because it could lead to a low contrast between pyelonephritis lesions and less-enhanced parenchyma during the poor perfusion state of the renal parenchyma on CEUS. In the missing case in the present study, the result of the DMSA scan showed multiple perfusion defects in the left kidney. Another case also showed diffusely increased parenchymal

echotexture, but no focal lesions were found on CEUS, and parenchymal damage due to VUR was confirmed in the follow-up ultrasound. The missed lesions on CEUS may be explained by the poor contrast between the true lesions and hypoperfused renal parenchyma. Another reason might be due to the operators' insufficient experience detecting the perfusion abnormality. The examinator will be able to see diffuse pathological changes only when they are familiar with the condition of normal renal perfusion in CEUS. It is necessary to accumulate more experience with diffuse renal perfusion abnormalities in the future.

This study has several limitations. First, most DMSA scans were performed when kidney lesions were detected on CEUS. Therefore, sensitivity and specificity of CEUS could not be assessed. It was difficult to recommend an additional examination when no abnormal findings were found on imaging tests in patients with febrile UTI. This is one reason for the lack of prospective CEUS studies for the diagnosis of APN in pediatric patients. Therefore, only local lesion concordance between DMSA scans, grayscale ultrasound, CDUS, and CEUS could be assessed. Second, not all DMSA scans were done in time [19]. In this study's five patients who underwent DMSA scans, the mean time from ultrasound examination to DMSA scan was 137.2 ± 166.6 days (2, 3, 6, 287, and 388 days, respectively). In the patient who underwent a DMSA scan after 388 days, there were no abnormal findings in the DMSA scan. However, renal scars do not appear in all APN patients [20], so APN may have resolved without sequelae through appropriate treatment in this patient. Therefore, the possibility of a false negative CEUS result cannot be ruled out. A follow-up ultrasound after CEUS (490.0 ± 168.5 days) was further investigated to overcome an insufficient number of DMSA scan results and to compensate for this study's limitation. We tried to improve the reliability of the negative results of CEUS by adding cases with normal findings on follow-up ultrasound. Additionally, a CEUS false-negative case was found in a patient with VUR, further adding an explanation of the false-negative risk during the CEUS examination. Third, the number of participants was small, even for a prospective study. Although the study was approved for use of ultrasound contrast agents, it was not easy to obtain consent from parents, so sufficient participants could not be secured. The clinical application of CEUS should be expanded based on safety evaluation data. There were no adverse events related to the use of ultrasonography contrast agents such as headache, nausea, or skin erythema in this study. In this study, the participants were infants at 9.0 months (2.0 months– 61.0 months). So, these data could be regarded as an additional value to the safety of ultrasound contrast agents.

In conclusion, this study showed that CEUS can reveal kidney perfusion defects in young infant patients with suspected APN without radiation exposure or sedation. CEUS may be a feasible and helpful technique compared to other ultrasound techniques for assessing the involvement of the kidney in febrile UTI.

## Author Contributions

**Conceptualization:** Han Byeol Lee, Seunghyun Lee, Young Hun Choi, Jung-Eun Cheon, Seul Bi Lee, Yeon Jin Cho, Yo Han Ahn, Seon Hee Lim.

**Data curation:** Han Byeol Lee, Seunghyun Lee, Young Hun Choi, Jung-Eun Cheon, Seul Bi Lee, Yeon Jin Cho, Yo Han Ahn, Seon Hee Lim.

**Formal analysis:** Han Byeol Lee, Seunghyun Lee, Young Hun Choi, Jung-Eun Cheon, Seul Bi Lee, Yeon Jin Cho, Yo Han Ahn, Seon Hee Lim.

**Funding acquisition:** Seunghyun Lee, Young Hun Choi.

**Supervision:** Seunghyun Lee, Young Hun Choi, Jung-Eun Cheon, Seul Bi Lee, Yeon Jin Cho, Yo Han Ahn, Seon Hee Lim.

**Writing – original draft:** Han Byeol Lee.

**Writing – review & editing:** Han Byeol Lee, Seunghyun Lee, Young Hun Choi, Jung-Eun Cheon, Seul Bi Lee, Yeon Jin Cho.

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
