## [Decision Letter · Decision Letter 0]

3 Jun 2022

PONE-D-22-08175Contrast-enhanced ultrasound for the diagnosis of acute pyelonephritis in pediatric patients with urinary tract infectionPLOS ONE

Dear Dr. Lee,

Thank you for submitting your manuscript to PLOS ONE. After careful consideration, we feel that it has merit but does not fully meet PLOS ONE’s publication criteria as it currently stands. Therefore, we invite you to submit a revised version of the manuscript that addresses the points raised during the review process.

We look forward to receiving your revised manuscript.

Kind regards,

Israel Franco, M.D.

Academic Editor

PLOS ONE

Journal Requirements:

2. In your statement, please include the full name of the IRB or ethics committee who approved or waived your study.

Additional Editor Comments :

Please see the comments from the reviewers and make appropriate revisions to the paper based on their comments.

if there are additional subjects that can be added to the paper please consider adding the information.

Reviewers' comments:

Reviewer's Responses to Questions

**Comments to the Author**

1. Is the manuscript technically sound, and do the data support the conclusions?

Reviewer #1: Yes

Reviewer #2: Partly

2. Has the statistical analysis been performed appropriately and rigorously? 

Reviewer #1: I Don't Know

Reviewer #2: Yes

3. Have the authors made all data underlying the findings in their manuscript fully available?

Reviewer #1: No

Reviewer #2: Yes

4. Is the manuscript presented in an intelligible fashion and written in standard English?

Reviewer #1: Yes

Reviewer #2: Yes

5. Review Comments to the Author

Reviewer #1: The authors attempt to show that CEUS can be used to diagnose acute pyelonephritis. We all would like to have minimal invasive imaging for our families. The mention most issues of their findings in their limitations. They do have a very small enrollment for a prospective study. While they tell us their inclusion and exclusion criteria they do not tell us the number of patient seen for a febrile in the same time period. As they mention it is also problematic that not all patients got a DMSA scan but rather a selected view. As a follow up study it would sure be interesting about the long term findings by possibly repeating the tests after >12 month. It would also be interesting if the patients had VUR.

In summary I do think the data is reportable with the limitations that they mention. The data is not enough to recommend CEUS alone but it is a start to possibly replace DMSA scans in the future.

Reviewer #2: CUES ultrasound represents an important avenue of future development for pediatric patients. The lack of radiation and sedation provide clear benefits to pediatric patients. While the study demonstrates concordance with DMSA scan in most patients, the sample size (5) does not allow for a strong assessment of specificity or sensitivity. The study provides further safety data in 21 pediatric patients given the absence of adverse events. The study adds evidence to safety and feasibility in pediatric patients.

6. PLOS authors have the option to publish the peer review history of their article (what does this mean?). If published, this will include your full peer review and any attached files.

Reviewer #1: No

Reviewer #2: No

---

## [Author Response · Author response to Decision Letter 0]

18 Jul 2022

Reviewer #1: The authors attempt to show that CEUS can be used to diagnose acute pyelonephritis. We all would like to have minimal invasive imaging for our families. The mention most issues of their findings in their limitations. They do have a very small enrollment for a prospective study. While they tell us their inclusion and exclusion criteria they do not tell us the number of patient seen for a febrile in the same time period. 

Thank you for your thoughtful comments. We fully agree with your comments.

A total of 72 patients with suspected febrile urinary tract infection underwent a renal ultrasound during the study period, of which 21 patients participated in this study. Because this study was approved only as a clinical trial for ultrasound contrast agent, it was difficult to obtain written consent from the subjects' parents, and as a result, the number of participants was small. However, the patients who participated in this study did not show any side effects, such as headache, nausea, or skin erythema, even though they were infants at 9.0 months of age (2.0 months – 61.0 months). Therefore, these data could be regarded as an additional value to the safety of ultrasound contrast agents. As you mentioned, the study is insufficient to recommend CEUS alone for the diagnosis of APN due to the small number of subjects; however, we think it will be useful for researchers interested in the role and safety of ultrasound contrast agents in the diagnosis of APN in pediatric UTI patients.

As they mention it is also problematic that not all patients got a DMSA scan but rather a selected view. As a follow up study it would sure be interesting about the long term findings by possibly repeating the tests after >12 month. It would also be interesting if the patients had VUR. In summary I do think the data is reportable with the limitations that they mention. The data is not enough to recommend CEUS alone but it is a start to possibly replace DMSA scans in the future.

Thank you for your thoughtful comments. We fully agree with your comments.

A total of six patients had a follow-up ultrasound instead of a DMSA scan after CEUS (490.0 ± 168.5 days), of which only one showed abnormal findings in the grayscale ultrasound, such as parenchymal atrophy. In this patient, the initial grayscale ultrasound showed diffusely increased parenchymal echotexture and slight pelvic dilatation of both kidneys, but no focal lesions were found on CEUS. A follow-up voiding cystourethrography in this patient revealed vesicoureteral reflux, and the parenchymal damage was confirmed in the follow-up ultrasound. Therefore, this case could be regarded as a false-negative case of CEUS.

As noted in the manuscript, CEUS may give false-negative results if there are diffuse renal parenchymal changes, as in this patient, and it might be due to the operators' insufficient experience detecting the perfusion abnormality. The examinator will be able to see diffuse pathological changes only when they are familiar with the condition of normal renal perfusion in CEUS. It is necessary to accumulate more experience with diffuse renal perfusion abnormalities in the future.

The remaining five patients showed normal ultrasound findings, which indirectly confirmed that there was no APN at the time of CEUS. A follow-up ultrasound was also performed on three patients who underwent a DMSA scan, and a follow-up ultrasound confirmed the atrophy or cortical thinning remained in the area where cortical defects were observed in the DMSA scan.

As you mentioned, it is problematic that not all patients got a DMSA scan, which leads to the selected view. A follow-up ultrasound after CEUS (490.0 ± 168.5 days) was further investigated to overcome an insufficient number of DMSA scan results and to compensate for this study's limitation. We tried to improve the reliability of the negative results of CEUS by adding cases with normal findings on follow-up ultrasound. Additionally, a CEUS false-negative case was found in a patient with VUR, further adding an explanation of the false-negative risk during the CEUS examination.

We revised and added the manuscript as follows:

Materials and Methods

Participants: A total of 72 patients with suspected febrile urinary tract infection underwent a renal ultrasound during the study period, of which 21 patients participated in this study.

Data Analysis: The same analysis of follow-up grayscale images was performed if the follow-up ultrasound was performed instead of a DMSA scan.

Results

A total of six patients had a follow-up ultrasound instead of a DMSA scan after CEUS (490.0 ± 168.5 days), of which only one showed abnormal findings in the grayscale ultrasound, such as parenchymal atrophy. In this patient, the initial grayscale ultrasound showed diffusely increased parenchymal echotexture and slight pelvic dilatation of both kidneys, but no focal lesions were found on CEUS. A follow-up voiding cystourethrography in this patient revealed vesicoureteral reflux, and the parenchymal damage was confirmed in the follow-up ultrasound. Therefore, this case could be regarded as a false-negative case of CEUS. The remaining five patients showed normal ultrasound findings, which indirectly confirmed that there was no APN at the time of CEUS. A follow-up ultrasound was also performed on three patients who underwent a DMSA scan, and a follow-up ultrasound confirmed the atrophy or cortical thinning remained in the area where cortical defects were observed in the DMSA scan.

Discussion

…

Another case also showed diffusely increased parenchymal echotexture, but no focal lesions were found on CEUS, and parenchymal damage due to VUR was confirmed in the follow-up ultrasound. The missed lesions on CEUS may be explained by the poor contrast between the true lesions and hypoperfused renal parenchyma.

…

Another reason might be due to the operators' insufficient experience detecting the perfusion abnormality. The examinator will be able to see diffuse pathological changes only when they are familiar with the condition of normal renal perfusion in CEUS. It is necessary to accumulate more experience with diffuse renal perfusion abnormalities in the future.

…

A follow-up ultrasound after CEUS (490.0 ± 168.5 days) was further investigated to overcome an insufficient number of DMSA scan results and to compensate for this study's limitation. We tried to improve the reliability of the negative results of CEUS by adding cases with normal findings on follow-up ultrasound. Additionally, a CEUS false-negative case was found in a patient with VUR, further adding an explanation of the false-negative risk during the CEUS examination.

…

In this study, the participants were infants at 9.0 months (2.0 months – 61.0 months). So, these data could be regarded as an additional value to the safety of ultrasound contrast agents.

Reviewer #2: CUES ultrasound represents an important avenue of future development for pediatric patients. The lack of radiation and sedation provide clear benefits to pediatric patients. While the study demonstrates concordance with DMSA scan in most patients, the sample size (5) does not allow for a strong assessment of specificity or sensitivity. The study provides further safety data in 21 pediatric patients given the absence of adverse events. The study adds evidence to safety and feasibility in pediatric patients.

Thank you for your thoughtful comments. We fully agree with your comments.

A total of 72 patients with suspected febrile urinary tract infection underwent a renal ultrasound during the study period, of which 21 patients participated in this study. Because this study was approved only as a clinical trial for ultrasound contrast agent, it was difficult to obtain written consent from the subjects' parents, and as a result, the number of participants was small. However, the patients who participated in this study did not show any side effects, such as headache, nausea, or skin erythema, even though they were infants at 9.0 months of age (2.0 months – 61.0 months). Therefore, these data could be regarded as an additional value to the safety of ultrasound contrast agents. As you mentioned, the study is insufficient to recommend CEUS alone for the diagnosis of APN due to the small number of subjects; however, we think it will be useful for researchers interested in the role and safety of ultrasound contrast agents in the diagnosis of APN in pediatric UTI patients.

A total of six patients had a follow-up ultrasound instead of a DMSA scan after CEUS (490.0 ± 168.5 days), of which only one showed abnormal findings in the grayscale ultrasound, such as parenchymal atrophy. In this patient, the initial grayscale ultrasound showed diffusely increased parenchymal echotexture and slight pelvic dilatation of both kidneys, but no focal lesions were found on CEUS. A follow-up voiding cystourethrography in this patient revealed vesicoureteral reflux, and the parenchymal damage was confirmed in the follow-up ultrasound. Therefore, this case could be regarded as a false-negative case of CEUS.

As noted in the manuscript, CEUS may give false-negative results if there are diffuse renal parenchymal changes, as in this patient, and it might be due to the operators' insufficient experience detecting the perfusion abnormality. The examinator will be able to see diffuse pathological changes only when they are familiar with the condition of normal renal perfusion in CEUS. It is necessary to accumulate more experience with diffuse renal perfusion abnormalities in the future.

The remaining five patients showed normal ultrasound findings, which indirectly confirmed that there was no APN at the time of CEUS. A follow-up ultrasound was also performed on three patients who underwent a DMSA scan, and a follow-up ultrasound confirmed the atrophy or cortical thinning remained in the area where cortical defects were observed in the DMSA scan.

As you mentioned, it is problematic that not all patients got a DMSA scan, which leads to the selected view. A follow-up ultrasound after CEUS (490.0 ± 168.5 days) was further investigated to overcome an insufficient number of DMSA scan results and to compensate for this study's limitation. We tried to improve the reliability of the negative results of CEUS by adding cases with normal findings on follow-up ultrasound. Additionally, a CEUS false-negative case was found in a patient with VUR, further adding an explanation of the false-negative risk during the CEUS examination.

We revised and added the manuscript as follows:

Materials and Methods

Participants: A total of 72 patients with suspected febrile urinary tract infection underwent a renal ultrasound during the study period, of which 21 patients participated in this study.

Data Analysis: The same analysis of follow-up grayscale images was performed if the follow-up ultrasound was performed instead of a DMSA scan.

Results

A total of six patients had a follow-up ultrasound instead of a DMSA scan after CEUS (490.0 ± 168.5 days), of which only one showed abnormal findings in the grayscale ultrasound, such as parenchymal atrophy. In this patient, the initial grayscale ultrasound showed diffusely increased parenchymal echotexture and slight pelvic dilatation of both kidneys, but no focal lesions were found on CEUS. A follow-up voiding cystourethrography in this patient revealed vesicoureteral reflux, and the parenchymal damage was confirmed in the follow-up ultrasound. Therefore, this case could be regarded as a false-negative case of CEUS. The remaining five patients showed normal ultrasound findings, which indirectly confirmed that there was no APN at the time of CEUS. A follow-up ultrasound was also performed on three patients who underwent a DMSA scan, and a follow-up ultrasound confirmed the atrophy or cortical thinning remained in the area where cortical defects were observed in the DMSA scan.

Discussion

…

Another case also showed diffusely increased parenchymal echotexture, but no focal lesions were found on CEUS, and parenchymal damage due to VUR was confirmed in the follow-up ultrasound. The missed lesions on CEUS may be explained by the poor contrast between the true lesions and hypoperfused renal parenchyma.

…

Another reason might be due to the operators' insufficient experience detecting the perfusion abnormality. The examinator will be able to see diffuse pathological changes only when they are familiar with the condition of normal renal perfusion in CEUS. It is necessary to accumulate more experience with diffuse renal perfusion abnormalities in the future.

…

A follow-up ultrasound after CEUS (490.0 ± 168.5 days) was further investigated to overcome an insufficient number of DMSA scan results and to compensate for this study's limitation. We tried to improve the reliability of the negative results of CEUS by adding cases with normal findings on follow-up ultrasound. Additionally, a CEUS false-negative case was found in a patient with VUR, further adding an explanation of the false-negative risk during the CEUS examination.

…

In this study, the participants were infants at 9.0 months (2.0 months – 61.0 months). So, these data could be regarded as an additional value to the safety of ultrasound contrast agents.

---

## [Decision Letter · Decision Letter 1]

18 Nov 2022

PONE-D-22-08175R1Contrast-enhanced ultrasound for the diagnosis of acute pyelonephritis in pediatric patients with urinary tract infectionPLOS ONE

Dear Dr. Lee,

Thank you for submitting your manuscript to PLOS ONE. After careful consideration, we feel that it has merit but does not fully meet PLOS ONE’s publication criteria as it currently stands. Therefore, we invite you to submit a revised version of the manuscript that addresses the points raised during the review process.

We look forward to receiving your revised manuscript.

Kind regards,

Shuai Ren

Academic Editor

PLOS ONE

Journal Requirements:

Additional Editor Comments (if provided):

The authors should address the concerns raised by reviewer # 3 and update the references as per request.

Reviewers' comments:

Reviewer's Responses to Questions

**Comments to the Author**

1. If the authors have adequately addressed your comments raised in a previous round of review and you feel that this manuscript is now acceptable for publication, you may indicate that here to bypass the “Comments to the Author” section, enter your conflict of interest statement in the “Confidential to Editor” section, and submit your "Accept" recommendation.

Reviewer #1: All comments have been addressed

Reviewer #3: All comments have been addressed

2. Is the manuscript technically sound, and do the data support the conclusions?

Reviewer #1: Partly

Reviewer #3: Partly

3. Has the statistical analysis been performed appropriately and rigorously? 

Reviewer #1: N/A

Reviewer #3: Yes

4. Have the authors made all data underlying the findings in their manuscript fully available?

Reviewer #1: Yes

Reviewer #3: Yes

5. Is the manuscript presented in an intelligible fashion and written in standard English?

Reviewer #1: Yes

Reviewer #3: Yes

6. Review Comments to the Author

Reviewer #1: They answered the questions according to their available data. Id they want to solidify their manuscript some=e additional data might be needed.

Reviewer #3: The authors sought to identify renal cortical changes associated with acute pyelonephritis (APN) related to vesicoureteral reflux.

A study such as this was done previously which showed contrast enhanced US to be woefully inaccurate by comparison to other modalities, such as DMSA, CT and MRI. (Majd M, Nussbaum Blask AR, Markle BM, Shalaby-Rana E, Pohl HG, Park JS, Chandra R, Rais-Bahrami K, Pandya N, Patel KM, Rushton HG. Acute pyelonephritis: comparison of diagnosis with 99mTc-DMSA, SPECT, spiral CT, MR imaging, and power Doppler US in an experimental pig model. Radiology. 2001 Jan;218(1):101-8. doi: 10.1148/radiology.218.1.r01ja37101. PMID: 11152787.) This study was performed with rigor. In all test subjects, UTI was strictly defined and confirmed, the presence of VUR was confirmed, and APN was confirmed by histology.

With this high bar set, the proposed manuscript leaves some questions.

First, only 40% of infants with APN have positive DMSA renal scans, so the inclusion of only 42 patients may not have been sufficiently powered to tell the difference between the two studies. It is suggestive that the kappa for agreement between CEUS and DMSA scan is 0.8; however, are we certain that these results are generalizable?

Secondly, another significant methodological problem in this paper is that DMSA scans were done 137 +/- 166 days following the suspected UTI presentation which might have been sufficient time for some APN lesions to resolve spontaneously.

I think the authors need to address these two concerns and updated the references to include prior work done.

7. PLOS authors have the option to publish the peer review history of their article (what does this mean?). If published, this will include your full peer review and any attached files.

Reviewer #1: No

Reviewer #3: No

---

## [Author Response · Author response to Decision Letter 1]

24 Jan 2023

Reviewer #1: They answered the questions according to their available data. Id they want to solidify their manuscript some additional data might be needed.

Thank you for your thoughtful comments. We chose the DMSA scan as a reference standard for comparison with contrast-enhanced ultrasound findings to detect the presence of APN. However, only a small number of DMSA results were included, so we added results with follow-up ultrasound through the revised manuscript.

Also, the intrinsic limitation of the US to detect the presence of APNs, the poor generalizability of diagnosing APNs with contrast-enhanced US, and the methodological problem of DMSA scans with varying times, which might be resolving lesions spontaneously, were raised and criticized. However, as you mentioned, we did our best to make as much evidence as possible based on our data, but more was needed to solve the problems raised.

Despite the limitations in this study, not only the typical APN findings seen in CEUS but also false-negative cases were included, which might be misdiagnosed in future clinical settings or research. The findings, such as the most visible time of the lesion and the safety of ultrasound contrast agents in infant patients, were also included and discussed. The results or cases in this study with young infant patients may be reference data for large-scale studies that will conduct CEUS testing in the future.

Therefore, we hope it will serve as primary or feasible data for future research on ultrasound examination using ultrasound contrast agents. In summary, we added and toned down the end of the manuscript title as “a feasibility test” considering the study's limitations and the hope for future research: “Title: Contrast-enhanced ultrasound for the diagnosis of acute pyelonephritis in pediatric patients with urinary tract infection: a feasibility study”.

Reviewer #3: The authors sought to identify renal cortical changes associated with acute pyelonephritis (APN) related to vesicoureteral reflux. A study such as this was done previously which showed contrast enhanced US to be woefully inaccurate by comparison to other modalities, such as DMSA, CT and MRI. (Majd M, Nussbaum Blask AR, Markle BM, Shalaby-Rana E, Pohl HG, Park JS, Chandra R, Rais-Bahrami K, Pandya N, Patel KM, Rushton HG. Acute pyelonephritis: comparison of diagnosis with 99mTc-DMSA, SPECT, spiral CT, MR imaging, and power Doppler US in an experimental pig model. Radiology. 2001 Jan;218(1):101-8. doi: 10.1148/radiology.218.1.r01ja37101. PMID: 11152787.) This study was performed with rigor. In all test subjects, UTI was strictly defined and confirmed, the presence of VUR was confirmed, and APN was confirmed by histology.

Thank you for your thoughtful comments. As you mentioned, DMSA scan is the best way to evaluate acute pyelonephritis and renal scarring [14, 15]. In the case of acute inflammation, the DMSA scan shows diminished uptake due to cortical vasoconstriction, obstruction of peritubular capillaries by inflammatory cells, and ischemia-related renal tubular dysfunction [15]. Renal scars also show cortical defects on DMSA scans due to permanent damage to the renal parenchyma [15].

The study you mentioned (Majd M et al., Radiology. 2001;218(1):101-8) was conducted in 2001 under rigorous conditions. At that time, there were many cases where APN could not be detected with grayscale ultrasound and color Doppler ultrasound technology (NOT contrast-enhanced ultrasound using ultrasound-specific contrast agent). The limitations of conventional ultrasound tools were the starting point of this study, so they were mentioned in the introduction and discussion of the manuscript.

CEUS is a new imaging tool with an ultrasound-specific contrast agent, Sonovue®, which was initially approved for the diagnostic imaging of liver tumors in adults and children by the FDA in the United States in 2016. Despite its off-label use, it is now increasingly used in European and Asian countries.

It can detect perfusion abnormalities by observing contrast enhancement in the renal parenchyma after injecting a microbubble contrast agent. Therefore, it can show hypoperfusion areas more objectively than other ultrasound modalities (e.g., grayscale, color Doppler, power Doppler) and is less sensitive to patient movement. In addition, no radiation exposure or sedation is required, and the ultrasound contrast agent used for the examination is safe, making it very useful for pediatric patients.

We added the references, including as study you mentioned (Majd M et al. Radiology. 2001 Jan:218(1):101-8).

14. Majd M, Nussbaum Blask AR, Markle BM, Shalaby-Rana E, Pohl HG, Park JS, Chandra R, Rais-Bahrami K, Pandya N, Patel KM, Rushton HG. Acute pyelonephritis: comparison of diagnosis with 99mTc-DMSA, SPECT, spiral CT, MR imaging, and power Doppler US in an experimental pig model. Radiology. 2001;218(1):101-8.

15. Ataei N, Madani A, Habibi R, Khorasani M. Evaluation of acute pyelonephritis with DMSA scans in children presenting after the age of 5 years. Pediatr Nephrol. 2005;20(10):1439-1444.

With this high bar set, the proposed manuscript leaves some questions.

First, only 40% of infants with APN have positive DMSA renal scans, so the inclusion of only 42 patients may not have been sufficiently powered to tell the difference between the two studies. It is suggestive that the kappa for agreement between CEUS and DMSA scan is 0.8; however, are we certain that these results are generalizable?

Thank you for your thoughtful comments. Through this study, we tried to evaluate the diagnostic ability of CEUS compared to the DMSA scan, but as you mentioned, not all DMSA scans were done in time, and the number of participants was small. Since both acute pyelonephritis and renal scarring show diminished uptake on the DMSA scan, the SNMMI guideline updated in 2022 recommends performing the test within two weeks of an acute infection episode when performing a DMSA scan to diagnose acute pyelonephritis [19]. We agreed with your opinion, but as mentioned in the Limitation section in our manuscript, it was difficult to recommend an additional examination when no abnormal findings were found on imaging tests in patients with febrile UTI. This is one reason for the lack of prospective CEUS studies for the diagnosis of APN in pediatric patients.

We also agree that the generalization of the study results is difficult due to the small number of participants. Although the study was approved for using ultrasound-specific contrast agents, obtaining consent from parents was difficult, so sufficient participants could not be secured. However, we did our best to make as much evidence as possible based on our data, such as the addition of the result of follow-up ultrasound, but more was needed to solve the problems raised.

Despite the limitations in this study, not only the typical APN findings seen in CEUS but also false-negative cases were included, which might be misdiagnosed in future clinical settings or research. The findings, such as the most visible time of the lesion and the safety of ultrasound contrast agents in infant patients, were also included and discussed. The results or cases in this study with young infant patients may be reference data for large-scale studies that will conduct CEUS testing in the future.

🡪 Therefore, we hope it will serve as primary or feasible data for future research on ultrasound examination using ultrasound contrast agents. In summary, we added and toned down the end of the manuscript title as “a feasibility test” considering the study's limitations and the hope for future research: “Title: Contrast-enhanced ultrasound for the diagnosis of acute pyelonephritis in pediatric patients with urinary tract infection: a feasibility study”.

Also, we added the following reference:

19. Vali, R., Armstrong, I.S., Bar-Sever, Z. et al. SNMMI procedure standard/EANM practice guideline on pediatric [99mTc]Tc-DMSA renal cortical scintigraphy: an update. Clin Transl Imaging 2022;10:173–184.

Secondly, another significant methodological problem in this paper is that DMSA scans were done 137 +/- 166 days following the suspected UTI presentation which might have been sufficient time for some APN lesions to resolve spontaneously.

Thank you for your thoughtful comments. In the five patients who underwent DMSA scans, the mean time from ultrasound examination to DMSA scan was 137.2 ± 166.6 days (2, 3, 6, 287, and 388 days, respectively). In the patient who underwent a DMSA scan after 287 days, there were multiple cortical defects in the right kidney, and contrast-enhanced CT taken at the same time revealed multifocal cortical thinning, indirectly suggesting the presence of prior APN. In another case, the patient who underwent a DMSA scan after 388 days, there were no abnormal findings in the DMSA scan, and a follow-up grayscale ultrasound also revealed no renal scarring.

However, as you mentioned, renal scars do not appear in all APN patients [20], so APN may have resolved without sequelae through appropriate treatment in this patient. Therefore, the possibility of a false negative CEUS result cannot be ruled out. For these reasons, follow-up ultrasound after CEUS (490.0 ± 168.5 days) was further investigated to overcome an insufficient number of DMSA scan results and to improve the reliability of the negative results of CEUS. Follow-up ultrasound was performed in 6 patients with negative CEUS results, and there was no evidence of renal scarring on follow-up ultrasound except for one patient with bilateral grade 5 VUR. Although evaluation is limited because follow-up ultrasound was not performed in all patients and renal scarring does not always occur in patients with APN, this may suggest that CEUS has a high negative predictive value for diagnosing APN in febrile UTI patients.

We added the Limitation section of the Discussion as following paragraph:

“Second, not all DMSA scans were done in time [19]. In this study's five patients who underwent DMSA scans, the mean time from ultrasound examination to DMSA scan was 137.2 ± 166.6 days (2, 3, 6, 287, and 388 days, respectively). In the patient who underwent a DMSA scan after 388 days, there were no abnormal findings in the DMSA scan. However, renal scars do not appear in all APN patients [20], so APN may have resolved without sequelae through appropriate treatment in this patient. Therefore, the possibility of a false negative CEUS result cannot be ruled out.”

19. Vali, R., Armstrong, I.S., Bar-Sever, Z. et al. SNMMI procedure standard/EANM practice guideline on pediatric [99mTc]Tc-DMSA renal cortical scintigraphy: an update. Clin Transl Imaging. 2022;10:173–184.

20. Faust WC, Diaz M, Pohl HG. Incidence of post-pyelonephritic renal scarring: a meta-analysis of the dimercapto-succinic acid literature. J Urol. 2009;181(1):290-298.

I think the authors need to address these two concerns and updated the references to include prior work done.

We really appreciate your constructive comments and additional suggestions. Despite these limitations, this study is meaningful because it is the first prospective study to evaluate the feasibility of CEUS. Also, there were no adverse events in 21 patients of infant age who underwent CEUS in this study, which may add evidence for the safety of ultrasound contrast agents in pediatric patients. CEUS is a test with many advantages and is expected to play an essential role in diagnosing acute pyelonephritis in pediatric patients in the future. 

Despite the limitations in this study, not only the typical APN findings seen in CEUS but also false-negative cases were included, which might be misdiagnosed in future clinical settings or research. The findings, such as the most visible time of the lesion and the safety of ultrasound contrast agents in infant patients, were also included and discussed. The results or cases in this study with young infant patients may be reference data for large-scale studies that will conduct CEUS testing in the future. 

We hope this study will help explain the feasibility and safety of CEUS in diagnosing APNs in pediatric patients.

---

## [Editor Report · Decision Letter 2]

22 Mar 2023

Contrast-enhanced ultrasound for the diagnosis of acute pyelonephritis in pediatric patients with urinary tract infection: a feasibility study

PONE-D-22-08175R2

Dear Dr. Lee,

We’re pleased to inform you that your manuscript has been judged scientifically suitable for publication and will be formally accepted for publication once it meets all outstanding technical requirements.

Kind regards,

Shuai Ren

Academic Editor

PLOS ONE

Additional Editor Comments (optional):

It could be accepted in this current form. Congratulations!
---

## [Editor Report · Acceptance letter]

28 Mar 2023

PONE-D-22-08175R2 

Contrast-enhanced ultrasound for the diagnosis of acute pyelonephritis in pediatric patients with urinary tract infection: a feasibility study 

Dear Dr. Lee:

I'm pleased to inform you that your manuscript has been deemed suitable for publication in PLOS ONE. Congratulations! Your manuscript is now with our production department. 

Kind regards, 

on behalf of

Dr. Shuai Ren 

Academic Editor

PLOS ONE